# Benefits of Conversion Surgery after Multimodal Treatment for Unresectable Pancreatic Ductal Adenocarcinoma

**DOI:** 10.3390/cancers12061428

**Published:** 2020-05-31

**Authors:** Hiroaki Yanagimoto, Sohei Satoi, Tomohisa Yamamoto, So Yamaki, Satoshi Hirooka, Masaya Kotsuka, Hironori Ryota, Mitsuaki Ishida, Yoichi Matsui, Mitsugu Sekimoto

**Affiliations:** 1Department of Surgery, Kansai Medical University, Hirakata 573-1010, Japan; yanagimh@med.kobe-u.ac.jp (H.Y.); tomot1226@yahoo.co.jp (T.Y.); yamakis@hirakata.kmu.ac.jp (S.Y.); ss_largehill@yahoo.co.jp (S.H.); kotsukam@hirakata.kmu.ac.jp (M.K.); ryoutahr@hirakata.kmu.ac.jp (H.R.); matsui@hirakata.kmu.ac.jp (Y.M.); sekimotm@hirakata.kmu.ac.jp (M.S.); 2Department of Pathology and Laboratory Medicine, Kansai Medical University, Hirakata 573-1010, Japan; mitsuaki.ishida@gmail.com

**Keywords:** unresectable pancreatic ductal adenocarcinoma, conversion surgery, early recurrence

## Abstract

Background: Traditionally, the treatment options for unresectable locally advanced (UR-LA) and metastatic (UR-M) pancreatic ductal adenocarcinoma (PDAC) are palliative chemotherapy or chemoradiotherapy. The benefits of surgery for such patients remains unknown. The present study investigated clinical outcomes of patients undergoing conversion surgery (CS) after chemo(radiation)therapy for initially UR-PDAC. Methods: We recruited patients with UR-PDAC who underwent chemo(radiation)therapy for initially UR-PDAC between April 2006 and September 2017. We analyzed resectability of CS, predictive parameters for overall survival, and early recurrence (within six months). Results: A total of 468 patients (108 with UR-LA and 360 with UR-M PDAC) were enrolled in this study, of whom, 17 (15.7%) with UR-LA and 15 (4.2%) with UR-M underwent CS. The median survival time (MST) and five-year survival of patients who underwent CS was 37.2 months and 34%, respectively; significantly better than non-resected patients (nine months and 1%, respectively, *p* < 0.0001). MST did not differ according to UR-LA or UR-M (50.5 vs. 29.0 months, respectively, *p* = 0.53). Early recurrence after CS occurred in eight patients (18.8%). Lymph node metastasis, positive washing cytology, large tumor size (>35 mm), and lack of postoperative adjuvant chemotherapy were statistically significant predictive factors for early recurrence. Moreover, the site of pancreatic lesion and administration of postoperative adjuvant chemotherapy were statistically significant prognostic factors for overall survival in the patients undergoing CS. Conclusion: Conversion surgery offers benefits in terms of increase survival for initially UR-PDAC for patients who responded favorably to chemo(radiation)therapy when combined with postoperative adjuvant chemotherapy.

## 1. Introduction

In the present-day situation, successful treatment of pancreatic ductal adenocarcinoma (PDAC) remains a therapeutic challenge, and the prognosis is generally poor [1]. Approximately 70% of patients with PDAC are not eligible for surgery, due to locally advanced or metastatic disease at the time of diagnosis [2]. Current guidelines of the National Comprehensive Cancer Network (NCCN) recommend nab-paclitaxel combined with gemcitabine (GnP) or FOLFIRINOX regimens as standard treatments for unresectable (UR) PDAC [3,4]. However, the median survival time (MST) for UR-PDAC remains low (9.2–13.5 months) [5,6,7]. The result of remarkable therapeutic response may occasionally become an indication for conversion surgery (CS) [8,9], which is defined as additional surgery for patients with UR-PDAC who responded favorably to multimodal treatment. However, the incidence and clinical effects are unknown at present. In the present study, we evaluated the clinical outcomes of CS after chemo(radiation)therapy for UR-PDAC, predictive parameters for early recurrence (within six months after CS) and prognostic parameters for overall survival (OS).

## 2. Patients and Methods

### 2.1. Study Population

This retrospective study was conducted by using data from a prospective database. We recruited all consecutive patients undergoing chemotherapy or chemoradiotherapy for UR-PDAC who were to the Department of Surgery, Kansai Medical University, for any treatment between April 2006 and September 2017. All patients were diagnosed with PDAC by cytology or pathology through endoscopic retrograde cholangiopancreatography or endoscopic ultrasound-guided fine-needle aspiration. We have previously reported the details of multidetector-raw computed tomography (MDCT) imaging for the diagnosis of PDAC and to rule out distant metastasis, as well as staging laparoscopy techniques [10,11]. Moreover, multimodal image findings such as contrast-enhanced ultrasonography (CE-US), gadoxetic acid–enhanced magnetic resonance imaging (EOB-MRI), and positron emission tomography (PET) were considered, and we certainly confirmed that all patients had UR-PC initially, according to the National Comprehensive Cancer Network (NCCN) guideline version 2.2017 [3,4].

Ethical approval: All procedures performed in studies involving human participants were in accordance with the ethical standards of the institutional research committee and with the 1964 Declaration of Helsinki and its later amendments or comparable ethical standards. Informed consent: Written informed consent was obtained from all study participants.

### 2.2. Data Collection

The following data were collected: clinicopathological characteristics, type of chemotherapy or chemoradiotherapy, frequency of CS, rates of peri-operative morbidity and mortality, predictive parameters for early recurrence (defined as within 6 months after CS), and prognostic parameters for OS.

### 2.3. Statistical Analysis

Data are presented as median (range). Continuous or categorical variables were compared by using the Mann–Whitney U, chi-square, or Fisher’s exact tests as appropriate. The OS and recurrence-free survival curves were estimated by using the Kaplan–Meier method and compared by using the log-rank test. Predictive factors identified by the univariate analysis were further examined by multivariate logistic regression analysis, to determine significant factors for OS and early recurrence among patients undergoing CS. The hazard ratio and 95% confidence intervals were calculated for all estimates. A two-tailed *p*-value of <0.05 was considered to be statistically significant. Calculations were performed by using JMP software, version 10 (SAS Inc., Cary, NC, USA).

## 3. Results

### 3.1. Patient Characteristics

Between April 2006 and September 2017, a total of 758 patients received treatment at our department; 290 of those patients underwent surgical resection. The remaining 468 patients with unresectable (UR) PDAC were finally enrolled in this study. Diagnoses were confirmed by using MDCT for 189 patients (40.4%) with unresectable locally advanced (UR-LA) PDAC and 279 (59.6%) with unresectable metastatic (UR-M) PDAC. We performed staging laparoscopy for 133 patients (28.4%) and palliative gastrojejunostomy for 20 patients (4.3%) with radiologically defined locally advanced disease. Positive peritoneal lavage cytology was identified in 30 patients (6.4%), peritoneal dissemination in 25 (5.3%), liver metastasis in 20 (4.3%), and other metastases in six (1.3%). In total, we treated 108 patients (23%) with UR-LA and 360 patients (77%) with UR-M (Figure 1).

Baseline characteristics of the study population and regimens that were selected as first-line treatment are listed in Table 1. The most frequently used regimen was gemcitabine (GEM), followed by GEM combined with S-1 and GEM combined with nab-paclitaxel.

The standard treatment for advanced pancreatic cancer has changed to gemcitabine since 2001, FOLFIRINOX since 2010, and GnP since 2013 in Japan. Moreover, gemcitabine combined with S-1 was often used as a treatment option. There was liver metastasis in 193 patients, peritoneal metastasis in 123 patients, and LA in 108 patients, respectively. Standardized regimen of chemotherapy in each time has been used in patients with UR-M PDAC. Patients with peritoneal metastasis were treated with S-1 + intravenous and intraperitoneal paclitaxel [12]. Moreover, we have implemented additional radiation therapy in UR-LA patients who still had the low-density area around celiac artery or superior mesenteric artery just before the planned conversion surgery for expecting the margin-negative resection. Positive peritoneal washing cytology was not defined as M1 at that time. Therefore, chemoradiation therapy was implemented for UR-LA with positive cytology.

### 3.2. Best Response After First-Line Treatment

Radiographic partial responses (PR) according to Response Evaluation Criteria in Solid Tumors (RECIST) criteria were observed in 45 patients (42%) with UR-LA and 86 (24%) with UR-M. Stable disease (SD) was observed in 38 patients (35%) with UR-LA and 119 (34%) with UR-M, and disease progression observed in 25 patients (23%) and 155 (42%), respectively. Disease control was achieved in 83 patients (77%) with UR-LA and 205 (58%) with UR-M. Furthermore, patients who could maintain PR or SD for more than eight months were shown in 44 patients (40.7%) with UR-LA and in 85 patients (23.6%) with UR-M, respectively.

### 3.3. Conversion Surgery

The major eligibility criteria for surgical exploration were as follows: clinical response (PR/CR) on CT imaging, reduction of tumor markers, fine performance status with patient’s willingness for surgery, and an interval of at least eight months since initial treatment [13]. In patients with peritoneal metastasis, disappearance of occult distant organ metastasis was confirmed by second-look staging laparoscopy in the context of the above criteria. In patients with liver metastasis, a maximum of three occult metastases on the liver surface were resected. In cases where tumor extension to the major vessels with attachment was observed, these patients were indicated for resection. Clinical staging and surgical exploration were re-evaluated at multidisciplinary team meetings.

During the study period, 36 patients were planned to undergo CS, and four underwent exploratory laparotomy for occult distant organ metastasis. Finally, CS was performed on 17 patients (15.7%) with UR-LA and 15 (4.2%) with UR-M. Some reasons were raised in 99 patients who had PR but did not undergo conversion surgery due to still UR-LA status on CT imaging and poor performance status. We performed subtotal stomach-preserving pancreaticoduodenectomy for 13 patients (40.6%), distal pancreatectomy for 11 (34.4%), total pancreatectomy for four (12.5%), and distal pancreatectomy with en-bloc celiac axis resection (DP-CAR) on four patients (12.5%) (Table 2). Concomitant CHA resection was done in four patients (12.5%), and concomitant portal vein resection was in 15 patients (46.9%). R0 resection was achieved in 29 patients (90.6%). The median operative time for the total study population was 441 (range 223–866) min, and the median intraoperative blood loss was 1250 (range 207–6301) mL. Although the complication of Clavien–Dindo classification ≥IIIa [14] was reported for eight patients (25.0%), there was no mortality. The median postoperative hospital stay was 14 (range 7–116) days. Histopathologically, Evans grade ≥III was noted in nine patients (28.1%), one of whom exhibited pathological complete response (pCR). The 23 patients (71.9%) received postoperative adjuvant chemotherapy; S-1 was administered to 13 patients (40.6%), GEM to three (9.4%), GEM plus S-1 to one (3.1%), and intraperitoneal infusion and intravenous administration of paclitaxel combined with S-1 to six (18.8%). Twenty-two patients (68.8%) completed adjuvant chemotherapy. The nine patients (28.1%) did not receive postoperative adjuvant chemotherapy, because of our policy of non-adjuvant chemotherapy in the first four patients, patient’s willingness (*n* = 3), or insufficient nutritional condition (*n* = 2).

### 3.4. Survival Analysis

The MST of the entire study population was 10 months, and the one- and two-year survival rates were 39% and 12%, respectively (Figure 2). Patients who achieved PR (*n* = 99) and did not undergo CS exhibited significantly increased survival in comparison with other patients (15 vs. 7.5 months, *p* < 0.0001; Figure 2). The MST following initial treatment of patients who underwent CS (*n* = 32) was 37.2 months, and the one-, three-, and five-year survival rates were 100%, 51%, and 34%, respectively. These patients also exhibited significantly increased survival than those who achieved PR (37.2 vs. 18 months, *p* < 0.0001; Figure 2).

When long PR/SD was defined as PR/SD persisting for eight months or more, survival was significantly better among patients who underwent CS compared with those with long PR/SD who did not undergo CS (*n* = 97) (37.2 vs. 19.5 months, *p* < 0.0001).

### 3.5. Comparison between Patients with Unresectable Locally Advanced and Metastatic Disease

Age, gender, tumor location, tumor diameter, tumor markers, pretreatment period to operation, postoperative complications, mortality, and length of hospital stay were not significantly different patients with UR-LA who underwent CS and those with UR-M who underwent CS (Table 2). Significant differences were identified in metastatic site and requirement of additional radiation therapy. There was no significant difference in survival from the time of initial treatment or from the time of CS between patients who underwent CS with UR-LA and those with UR-M (50.5 vs. 29.0 months, *p* = 0.53; 25.0 vs. 21.0 months, *p* = 0.61, respectively; Figure 3).

### 3.6. Recurrence-Free Survival

The MST from CS was 23 months, and the median recurrence-free survival time was 13 months (Figure 4).

Recurrence was confirmed in 20 (62.5%) of 32 patients who underwent CS, presenting as peritoneal dissemination in seven patients, locoregional recurrence in six, liver metastasis in five, and lung metastasis in two. Recurrence within six months after CS was observed in six patients (18.8%), presenting as liver metastasis in three patients, peritoneal dissemination in two, and local recurrence in one. One of those patients received GEM, and four patients received S-1 as adjuvant chemotherapy after CS. After relapse was confirmed, two of the six patients received the same regimen as was administered for initial treatment; these patients survived 23 and 32 months after CS. Patients who suffered recurrence within six months after CS had relatively poorer prognoses than non-recurrent patients (25.5 vs. 50.5 months, *p* = 0.22). Multivariate logistic regression analyses revealed that lymph node metastasis, washing cytology positive, large tumor (>35 mm), and lack of postoperative adjuvant chemotherapy were predictive factors for early recurrence (Table 3).

### 3.7. Prognostic Factors for Overall Survival Among Patients Who Underwent Conversion Surgery

The multivariate analysis revealed the site of pancreatic lesion and postoperative adjuvant chemotherapy to be statistically significant prognostic factors for OS among patients undergoing CS (*p* = 0.0092 and *p* < 0.0001, respectively). Other parameters, including reduction of tumor markers and Evans grading, were not significantly risk factors (Table 4).

## 4. Discussion

Despite recent advances in diagnostic medicine, detection of pancreatic cancer while it is within the resectable stage remains a clinical challenge. According to systematic reviews, the condition is not detected until it has reached the locally advanced or metastatic stage in 30–40% and 40–50% of patients, respectively [15,16,17]. Thus, despite the development of chemotherapy, the prognosis of patients with UR-PDAC remains poor, with a median survival of 9.2–13.5 months and low rates of long-term survival. [5,6,7]

Favorable outcomes may be achieved for a certain period of time, through the use of chemo(radiation)therapy for patients with unresectable malignancies, and this treatment can be converted to surgical resection, as required. Conversion surgery represents a new therapeutic strategy which may improve short- and long-term outcomes of patients with UR-PDAC. Several articles have reported the utility of CS in such patients, as well as the positive effects on prognosis [17,18,19,20,21,22,23,24,25,26]. In the present study, the rate of CS among patients with UR-LA and UR-M was similar to that reported previously [25]. We found the long-term prognosis; one-, three-, and five-year OS rates from initial treatment; and MST were significantly better among patients with long PR/SD who did not undergo CS, although there were no significant differences in survival with relation to UR-LA or UR-M. Therefore, CS should be considered even for patients initially diagnosed with UR-M if they exhibit surgical indicators. Considering the favorable long-term survival of patients who underwent CS in the present study, our suggestion of tumor extension to the major vessels with attachment as an indication for surgery appears reasonable. However, early recurrence was observed in almost 20% of patients, in line with the findings of Wright et al., who reported that seven out of 23 patients (30.4%) with metastatic PDAC who underwent CS experienced early recurrence. Other studies have also reported early recurrence rates after conversion surgery of approximately 30% [27,28,29]. This would suggest that patients cannot be expected to survive longer than patients who receive non-surgical treatment, and conversion surgery may be harmful to patients because of the high risk of mortality and morbidity associated with extensive pancreatectomy. The early recurrence rate should be decreased as much as possible for patients undergoing CS [30]. Thus, although CS can prolong OS, early recurrence remains a considerable risk. Appropriate preoperative selection of patients for CS is absolutely necessary in order to improve prognosis. The relatively strict surgical indication employed in the present study resulted in prolonged survival and a reduced incidence of early recurrence. In contrast, a review article reported that some authors recommend patients with UR-PDAC who did not experience progression after chemo(radiation) therapy should be offered surgical exploration [30]. The resectability and MST of patients in these studies who underwent CS ranged from 20% to 69% (median, 52%) and from 19.5 to 33 months (median, 21.9 months), respectively. Strict criteria may lead to lower resectability but longer OS, as a result of patient selection. Broad criteria may be associated with higher resectability but shorter OS, due to the risk of early recurrence after conversion surgery. Surgical indications for CS should be carefully decided through discussion in a multidisciplinary meeting.

To the best of our knowledge, there have been no previous studies on predictive factors of early recurrence after CS. The present study demonstrates that lymph node metastasis, positive washing cytology, large tumor size (>35 mm), and lack of postoperative adjuvant chemotherapy are significant predictive factors for early recurrence after CS. Thus, tumor size and washing cytology may be important preoperative factors which should be considered during patient selection for CS. Staging laparoscopy should be routinely performed before proceeding with CS in order to exclude patients with positive washing cytology. Metastatic site, decreased CA19-9 level, and performance status are not significant predictive factors for early recurrence. Several articles have reported that decreased CA19-9 levels after multimodal therapy represent a reliable predictive factor for resectability, OS, and DFS [21,29,30,31,32,33,34,35]. In most patients of the present study, CA19-9 decreased to within normal limits after multimodal treatment. Although the optimal selection criteria for surgical exploration or resection remain controversial for patients with initially UR-PDAC, it may be appropriate to base decision-making for CS on clinical response (defined by RECIST criteria) and decreased CA19-9 level after multimodal therapy [30].

Regarding pathological examination, the utility of Evans classification reflecting the extent of tumor degeneration or necrosis has been extensively studied as a prognostic factor after preoperative treatment [24,35,36,37,38]. There have been reports of the association between histopathological responses to chemo(radiation)therapy and the prognosis of patients with PDAC [24,35,36,37,38]. Chaterjee et al [36]. reported that 42 (18.8%) of 223 patients with resectable PDAC who received neoadjuvant chemotherapy were classified as Evans grade ≥III and had better survival rates than patients classified as Evans grade <III. Moreover, White et al. [37] suggested histologic response to be a useful surrogate marker for treatment efficacy, but Evans grade was not found to be a prognostic factor of CS in the present study.

The present study has some limitations which should be acknowledged. Firstly, it is a single-institute and retrospective study involving a small number of patients. All studies on this subject, to date, are retrospective studies, and so we believe that a prospective study is necessary to define the efficacy of CS. In Japan, the results of the PREP-04 trial (UMIN000017793)—a multi-institutional prospective cohort study investigating clinical outcomes of CS on patients with initially UR-PDAC— will be published in the near future. Given that only patients who responded favorably to chemo(radiation)therapy were analyzed among all patients with UR-PDAC, a selection bias exists. The development of an effective therapeutic strategy involving combined multimodal treatment with surgical resection is critical.

## 5. Conclusions

In conclusion, CS can provide clinical benefits, including increased survival for patients with initially UR-PDAC who have responded favorably to chemo(radiation)therapy. In addition to CS, postoperative adjuvant chemotherapy is necessary to prolong survival. It is essential that efforts are made to reduce early recurrence and to investigate surrogate markers in order to determine appropriate indications for surgery.

## Figures and Tables

**Figure 1 cancers-12-01428-f001:**
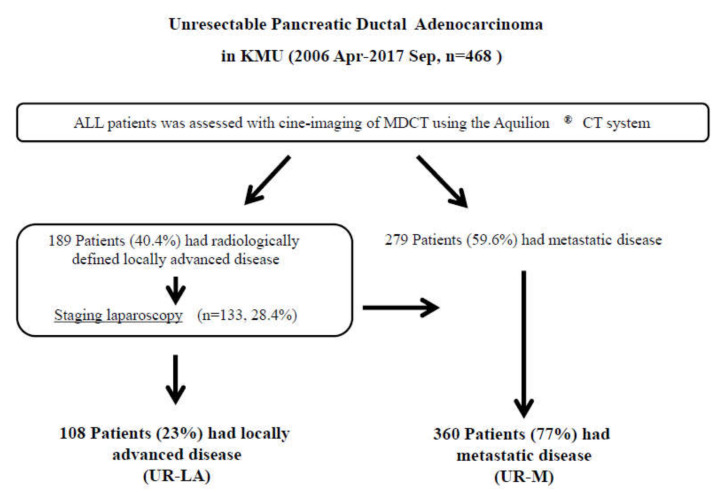
Study flow diagram. Diagnoses were made, using multidetector-raw computed tomography. In total, 189 patients were diagnosed with unresectable locally advanced pancreatic ductal adenocarcinoma (UR-LA PDAC), and 279 patients were diagnosed with unresectable metastatic (UR-M) PDAC. We performed staging laparoscopy for 133 patients with radiologically defined locally advanced disease. We finally enrolled 108 patients with UR-LA PDAC and 360 patients with UR-M PDAC in the present study. Abbreviations: KMU, Kansai Medical University; MDCT, multidetector-raw computed tomography; UR-LA, unresectable locally advanced; UR-M, unresectable metastatic.

**Figure 2 cancers-12-01428-f002:**
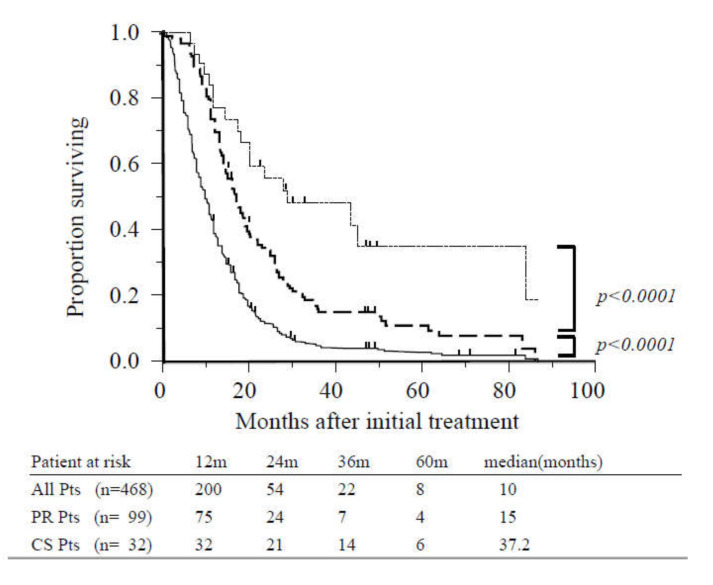
Overall survival of all patients, patients with radiographic partial response, and patients who underwent conversion surgery. The median overall survival (OS) for the study population (solid line, *n* = 468) was 10 months. Survival was significantly better among patients with partial response (dashed line, *n* = 99) compared with other cases (*p* < 0.0001). Survival of patients who underwent conversion surgery (dotted line, *n* = 32) was significantly better than those with partial response (*p* < 0.0001). Abbreviations: CS, conversion surgery; Pts, patients; PR, partial response.

**Figure 3 cancers-12-01428-f003:**
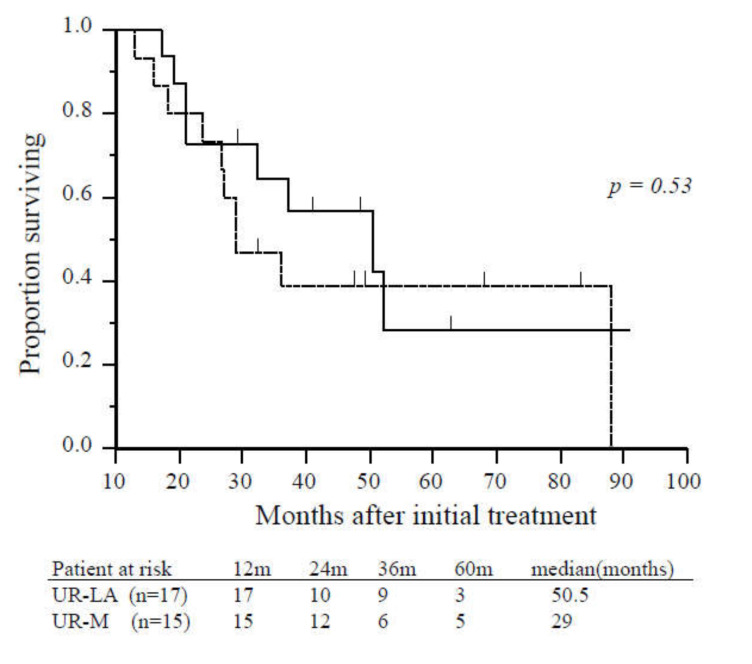
Overall survival of patients with unresectable locally advanced or metastatic disease who underwent conversion surgery. There was no significant difference in overall survival between patients with unresectable locally advanced (solid line, *n* = 17) and unresectable-metastatic disease (dashed line, *n* = 15) (*p* = 0.53). Abbreviations: UR-LA, unresectable locally advanced; UR-M, unresectable metastatic.

**Figure 4 cancers-12-01428-f004:**
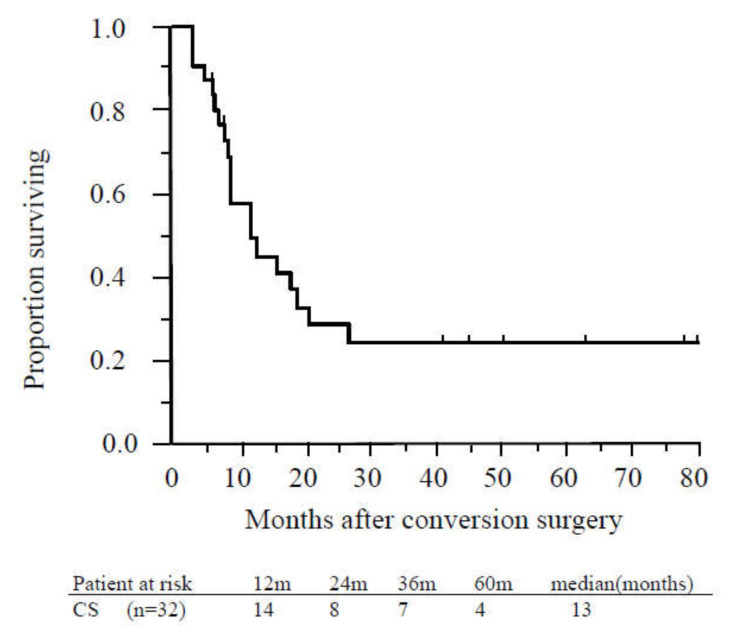
Recurrence-free survival of patients who underwent conversion surgery. The median recurrence-free survival time of patients who underwent conversion surgery was 13 months. Abbreviations: CS, conversion surgery.

**Table 1 cancers-12-01428-t001:** Baseline patient characteristics.

Variables	UR-LA (*n* = 108)	UR-M (*n* = 360)	*p*-Value
Age (years), median (range)	69(38–84)	67(33–86)	0.15
Male/female, *n* (%)	54(50)/54(50)	210(58.3)/150(41.7)	0.12
ECOG PS, *n* (%): 0/1/2	86(79.6)/18(16.6)/4(3.8)	218(60.5)/127(35.2)/15(4.2)	0.0004
Tumor location, *n* (%)			
Head/Body-tail	73(67.6)/35(32.4)	148(41.1)/212(58.9)	<0.0001
Tumor size (mm)	38(20-76)	40(15-83)	0.062
CA19-9 (U/L)	237(1.1-8949)	580(1-12219)	<0.0001
Extent of disease, *n* (%)			
Localized			
Metastatic site	108(100)		
Liver		193(53.6)	
Peritoneum		123(34.2)	
Lung/LN/Other		15(4.2)/23(6.4)/6(1.6)	
Treatment, *n* (%)			
GEM	17(15.7)	125(34.7)	
GEM + Erlotinib	2(1.9)	14(3.9)	
S-1	5(4.6)	38(10.6)	
GS	31(28.7)	42(11.6)	
GnP	13(12.0)	63(17.5)	
FOLFIRINOX	12(11.1)	21(5.8)	
S-1or GnP or GS plus PTX (i.p. + i.v.)	0(0)	43(11.9)	
Chemoradiotherapy	27(25)	10(2.8)	
Other	1(0.9)	4(1.1)	

UR-LA: unresectable locally advanced pancreatic cancer, UR-M: metastatic pancreatic cancer, PS: performance status, GEM: gemcitabine, GS: S-1 combined with gemcitabine, GnP: nab-paclitaxel combined with gemcitabine, PTX: paclitaxel.

**Table 2 cancers-12-01428-t002:** Patient characteristics of conversion surgery.

Variables	*n* = 32	UR-LA (*n* = 17)	UR-M (*n* = 15)	*p*-Value
Age(years), median (range)	66 (36–84)	65 (38–75)	69 (50–83)	0.135
Male: Female, *n* (%)	14 (44): 18 (56)	7(41):10(59)	7(47):8(53)	0.754
Ph: Pbt, *n* (%)	14 (44): 18 (56)	9 (53): 8 (47)	5 (33): 10 (67)	0.264
Tumor Size(mm), median (range)	36 (25–74)	35 (25–55)	40 (27–74)	0.747
Mets site: None:L:P, *n* (%)	17 (53): 4 (13):11 (34)	17 (100): 0 (0):0 (0)	0(0): 4 (27):11 (73)	<0.0001
CA19-9; U/mL, median (range)	278 (1.2–3400)	126 (8.4–2200)	984 (6.6–1953)	0.209
Preoperative CA19-9	29.1(1.0–181.9)	39.7(1.0–181.9)	19(1.0–73.9)	0.42
Primary Treatment				
GEM or GS	7	4	3	
GEM + nab-PTX (GnP)	7	4	3	
S1 or GEM based + ip PTX	8	0	8	
FOLFIRINOX	3	3	0	
GEM or S-1 or GS + RT (50.4 Gy)	7	6	1	
Radiation, *n* (%)	13(41)	11(65)	2(13)	0.002
Pretreatment period to op; (median, range)	9.5(4–28)	10 (4–28)	9 (6–16)	0.6207
RECIST (CR: PR), *n* (%)	1 (3.1%): 31(96.9%)	0(0):17(100)	1(7):14(63)	0.153
Operative time(min)	454(223–866)	441(223–655)	467(227–866)	0.36
Intraoperative blood loss(mL)	1229(207–6301)	1087(237–2931)	1255(207–6301)	0.58
Blood transfusion (U)	0(0–12)	0(0–7)	0(0–12)	0.42
PD: DP: DP-CAR: TP, *n* (%)	13 (40): 11(34): 4(13): 4(13)	9(52):4(24):2(12):2(12)	5(33):7(47):2(13):1(7)	0.257
-CHA/CA/PV resection-	-3(9)/4(13)/15(47)-	-3(18)/2(12)/9(52)-	-0(0)/2(13)/6(40)-	
Residual tumor (R0: R1), *n* (%)	29(91): 3(9)	16(94):1(6)	13(87):2(13)	0.471
Postop comp/Mortality (%)	8(25)/0(0)	2(12)/0(0)	6(40)/0(0)	0.066/0
Hospital stay (median, range)	14 (7–114)	11 (7–41)	14 (7–114)	0.271
Evans (I/IIa/IIb/III/IV, (%))	1(3)/12(38)/10(31)/8(25)/1 (3)	1(6)/7(41)/5(29)/4(24)/0 (0)	0(0)/5(33)/5(33)/4 (27)/1(7)	0.695

Ph: pancreas head, Pbt: pancreas body and tail, Mets: metastasis, L: liver, P: peritoneum, GEM: gemcitabine, GS: S-1 combined with gemcitabine, GnP: nab-paclitaxel combined with gemcitabine, PTX: paclitaxel, RT: radiation, PD: pancreaticoduodenectomy, DP: distal pancreatectomy, DP-CAR: distal pancreatectomy with en-bloc celiac axis resection, CHA: common hepatic artery, CA: celiac artery, PV: portal vein.

**Table 3 cancers-12-01428-t003:** Predictive factor for the recurrence within six months after CS (Univariate and multivariate logistic regression analyses).

Variables	Univariate Analysis		*p*-Value	Multivariate Analysis		*p*-Value
	HR	95% CI		HR	95% CI	
UR-M vs. UR-LA	1.16	0.19–7.37	0.87			0.07
Pbt vs. Ph	4.99	0.68–102	0.12			0.07
Tumor Size (>35 mm vs. <35 mm)	5.83	0–0.40	0.007	2.16	0–2.31	0.003
Pretreatment period (<8 m vs. >8 m)	2.22	0.29–46.22	0.47			0.38
Reduction of CA19-9 or DUPAN-2 (<70% vs. >70%)	5.99	0.80–50.84	0.08			0.38
LN mets (+) vs. (−)	4.29	0.58–88.2	0.16	4.5	0.40–11.10	0.01
R0 vs. R1	3.11	0–4.00	0.25			0.99
CY (+) vs. (−)	2.74	0.31–20.3	0.34	1.11	0.56–1.70	<0.0001
Evans I-IIa vs. IIb-IV	1.47	0.24–11.9	0.68			0.46
Adjuvant Tx (−) vs. (+)	1.36	0.16–8.74	0.76	2.96	0.32–3.06	0.0029

CS: conversion surgery, HR: hazard ratio, CI: confidential interval, LN: lymph node, R: residual tumor, CY: washing cytology, Tx: chemotherapy.

**Table 4 cancers-12-01428-t004:** Univariate and multivariate analysis of prognostic factor of overall survival in CS group.

Variables	Univariate Analysis		*p*-Value	Multivariate Analysis		*p*-Value
	HR	95% CI		HR	95% CI	
UR-M vs. UR-LA	1.34	0.53–3.54	0.53			0.44
Pbt vs. Ph	1.24	0.49–3.40	0.65	14.14	1.86–182	0.0092
Tumor Size (>35 mm vs. <35 mm)	2.47	0.89–7.53	0.08			0.61
Pretreatment period (<8 m vs. >8 m)	0.79	0.22–2.25	0.68			0.27
Reduction of CA19-9 or DUPAN-2 (<70% vs. >70%)	1.08	0.38–3.89	0.89			0.086
LN mets (+) vs. (−)	1.07	0.38–2.86	0.89			0.5
R1 vs. R0	1.76	0.27–6.41	0.49			0.19
CY (+) vs. (−)	2.91	0.98–7.71	0.05			0.08
Evans I-IIa vs. IIb-IV	1.77	0.65–4.71	0.26			0.05
Adjuvant Tx (−) vs. (+)	4.63	1.76–12.13	0.0024	367.22	20.16–15093	<0.0001

CS: conversion surgery, HR: hazard ratio, CI: confidential interval, LN: lymph node, R: residual tumor, CY: washing cytology, Tx: chemotherapy.

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
