# Peer review of "Benefits of Conversion Surgery after Multimodal Treatment for Unresectable Pancreatic Ductal Adenocarcinoma"

_cancers, 2020, doi:10.3390/cancers12061428_

Round 1

Reviewer 1 Report

This study investigates the effect of neoadjuvant therapy and following conversion surgery for patients with unrersectable PDAC -either unresectable locally or due to metastases. The authorscan Show that patients who receive conversion surgery have a significant benefit. Although many studies have reported on this issue in recent years, there are several aspects that make the work different: the fact that adjuvant therapy was given to many patients (72%) after neoadjuvant Treatment and surgery and that this improved the prognosis which is an uclear Topic to date and the identification of prognostic facort to predict early recurrence - an alos rarly addressed Topic in this Setting.

Comments:

The criteria for selection for conversion surgery should be discussed. the use of conventional cross-sectional Imaging often Fails in Terms of Response Evaluation, still it seems to be the most important Feature in the authors' Institution to consider Exploration. Don't the authors think that a much hjigher number of patients could have undergone Exploration if only stable disease (not partial Response or more) had been the entry criterion? from my experience, many Tumors look unchange after neoadjuvant TX but still are resectable as only scar tissue is found at the critical margins.

Can the authors explain their Standards to Chose a specific neoadjuvant protocol (of the 8 protcols that are mentioned)? this seems to be done rather unclear - especially as patients with metastatic disease received chemoradiation (n=10). What was the rationale for this?

Why did 43 metastatic patients receive intraperitoneal chemotherapy (Assumed that iv+ip means intravenous+intraperitoneal, table 1)? How was this done? once or continously via drainages?

Lines in table 1 have shifted, should be corrected.

Paragraph 5. (conclusion) is missing.

Author Response

Thank you very much indeed for your valuable comments for making this paper better.

Comments

  1. 1. The criteria for selection for conversion surgery should be discussed. the use of conventional cross-sectional Imaging often Fails in Terms of Response Evaluation, still it seems to be the most important Feature in the authors' Institution to consider Expl Don't the authors think that a much hjigher number of patients could have undergone Exploration if only stable disease (not partial Response or more) had been the entry criterion? from my experience, many Tumors look unchange after neoadjuvant TX but still are resectable as only scar tissue is found at the critical margins.

Ans: As the reviewer commented, many tumors look unchanged after neoadjuvant treatment and some reports showed the indication of surgical exploration if only stable disease after neoadjuvant treatment. We have the policy of relatively strict surgical indication for conversion surgery which is tumor shrinkage, reduction of tumor marker, fine performance status and good nutrition. Because we believe that effectiveness of conversion surgery in UR-PDAC who responded favorably to neoadjuvant treatment depends upon whether it is done safely, allowing patients to return to productive lives, with the improved postoperative life expectancy.

The following sentences were added on line 2, page17 in Discussion

“Strict criteria may lead to lower resectability but longer OS as a result of patient selection. Broad criteria may be associated with higher resectability but shorter OS due to the risk of early recurrence after conversion surgery.”

  1. Can the authors explain their Standards to Chose a specific neoadjuvant protocol (of the 8 protcols that are mentioned)? this seems to be done rather unclear - especially as patients with metastatic disease received chemoradiation (n=10). What was the rationale for this?

Ans: I appreciate for your good question. The following sentences and a reference12 (Satoi et al. Ann Surg. 2017) were added on line 16, page 8 in Results (3.1 Patient characteristics.)

“The standard treatment for advanced pancreatic cancer has changed to gemcitabine since 2001, FOLFIRINOX since 2010, and GnP since 2013 in Japan. Moreover, gemcitabine combined with S-1 was often used as treatment option. There was liver metastasis in 193 patients, peritoneal metastasis in 123 patients and LA in 108 patients, resepectively. Standardized regimen of chemotherapy in each time has been used in patients with UR-M PDAC. Patients with peritoneal metastasis was treated with S-1 + intravenous and intraperitoneal paclitaxel12. Moreover, we have implemented additional radiation therapy in UR-LA patients who still had the low density area around celiac artery or superior mesenteric artery just before the planned conversion surgery for expecting the margin-negative resection. Positive peritoneal washing cytology was not defined as M1 at that time. Therefore, chemoradiation therapy was implemented for UR-LA with positive cytology.”

  1. Why did 43 metastatic patients receive intraperitoneal chemotherapy (Assumed that iv+ip means intravenous+intraperitoneal, table 1)? How was this done? once or continously via drainages?

Ans: We have developed the regimen of paclitaxel intravenous +intraperitoneal infusion and reported extremely effective treatment for the patients with peritoneal dissemination (Satoi et al. Ann Surg. 2017). A peritoneal access port was implanted in the lower abdomen, with a catheter placed in the pelvic cavity. Paclitaxel was repeatedly administered intraperitoneally 20 mg / m2 on days 1 and 8 at outpatient unit.

  1. Lines in table 1 have shifted, should be corrected.

Ans: As you mention, I corrected the table 1.

  1. Paragraph 5. (conclusion) is missing.

Ans: As you mention, I added the conclusion in the revised manuscript.

I apologize for our mistake.

The following sentences that "Written informed consent was obtained from all study participants" and “This study and all its protocols were approved by the institutional review board of our hospital (Protocol No. H150936)” were mistake. It was the retrospective study, and we provided Information disclosure documents. Informed consent for conversion surgery but not for clinical research was obtained from all patients.

I would be pleased if you could accept that. We would remake it in the text.

Any criticism or comments regarding the suitability of this paper for publication would be greatly appreciated. Thank you very much for your time.

Yours sincerely

Hiroaki Yanagimoto, M.D., Ph.D

Department of Surgery,

Kansai Medical University

Reviewer 2 Report

This is an important study looking for a subset of patients who may benefit from surgery after receiving chemo(radiation) therapy. The authors follow unresectable patients from their institution that had started on chemo(radiation). A small number of these patients met the criteria for surgery (including a PR or CR). Patients who underwent conversion surgery outperformed those who had a PR but did not undergo surgery. It is unclear why the 99 patients who had a PR but did not have conversion surgery, did not undergo surgery. The fact that they were not able to have surgery could be the underlying reason why they had a "poorer" performance. In theory, the same "type" of patients would undergo surgery, but it would be unethical to perform a randomized study in this manner. 

Additionally, it would be interesting to compare the recurrence free rate of those who were unresectable and received conversion therapy plus adjuvant therapy compared to those who were resectable at diagnosis. 

Other minor comments: 

  1. In the abstract, methods section, line 18- clarify how the patients were followed after initial recruitment. It's unclear that the patients were followed after they had chemo(radiation) therapy.

     2. Figure 1 appears before Table 1 in the text but Table 1 is displayed after           Figure. 

    3. The numbering for Figure 3 and 4 is out of order. 

Author Response

Thank you very much indeed for your valuable comments for making this paper better.

Comments

This is an important study looking for a subset of patients who may benefit from surgery after receiving chemo(radiation) therapy. The authors follow unresectable patients from their institution that had started on chemo(radiation). A small number of these patients met the criteria for surgery (including a PR or CR). Patients who underwent conversion surgery outperformed those who had a PR but did not undergo surgery. It is unclear why the 99 patients who had a PR but did not have conversion surgery, did not undergo surgery. The fact that they were not able to have surgery could be the underlying reason why they had a "poorer" performance. In theory, the same "type" of patients would undergo surgery, but it would be unethical to perform a randomized study in this manner.

Ans: I appreciate for your good question. We have the policy of relatively strict surgical indication for conversion surgery which is tumor shrinkage, reduction of tumor marker, fine performance status and good nutrition. Because we believe that effectiveness of conversion surgery in UR-PDAC who responded favorably to neoadjuvant treatment depends upon whether it is done safely, allowing patients to return to productive lives, with the improved postoperative life expectancy.

The following sentences were added on line 13, page10 in Results (3.3 Conversion surgery)

“Some reasons were raised in 99 patients who had PR but did not undergo conversion surgery due to still UR-LA status on CT imaging and poor performance status.”

I also think that it would be difficult to perform to the randomized study.

Additionally, it would be interesting to compare the recurrence free rate of those who were unresectable and received conversion therapy plus adjuvant therapy compared to those who were resectable at diagnosis.

Ans: That is also very interesting, but we have no data to compare recurrence free rate at that time.

Other minor comments:

1.In the abstract, methods section, line 18- clarify how the patients were followed after initial recruitment. It's unclear that the patients were followed after they had chemo(radiation) therapy.

Ans: Most patients were followed at outpatient unit or palliative care unit in our hospital.

  1. Figure 1 appears before Table 1 in the text but Table 1 is displayed after Figure.

Ans: I will change Figure 1 to display before Table 1.

  1. The numbering for Figure 3 and 4 is out of order.

Ans: As you mention, I will change correctly.

I apologize for our mistake.

The following sentences that "Written informed consent was obtained from all study participants" and “This study and all its protocols were approved by the institutional review board of our hospital (Protocol No. H150936)” were mistake. It was the retrospective study, and we provided Information disclosure documents. Informed consent for conversion surgery but not for clinical research was obtained from all patients.

I would be pleased if you could accept that. We would remake it in the text.

Any criticism or comments regarding the suitability of this paper for publication would be greatly appreciated. Thank you very much for your time.

Yours sincerely

Hiroaki Yanagimoto, M.D., Ph.D

Department of Surgery,

Kansai Medical University